# Repurposing 9-Aminoacridine as an Adjuvant Enhances the Antimicrobial Effects of Rifampin against Multidrug-Resistant *Klebsiella pneumoniae*

Pengfei She,[a] Yimin Li,[a] Zehao Li,[a] Shasha Liu,[a] Yifan Yang,[a] Linhui Li,[a] Linying Zhou,[b] Yong Wu[b]

[a]Department of Laboratory Medicine, The Third Xiangya Hospital of Central South University, Changsha, China
[b]Department of Laboratory Medicine, The Affiliated Changsha Hospital of Xiangya School of Medicine, Central South University, Changsha, China

Pengfei She and Yimin Li contributed equally to this work and share first authorship. Author order was determined in order of increasing seniority.

**ABSTRACT** The increasing occurrence of extensively drug-resistant and pan-drug-resistant *K. pneumoniae* has posed a serious threat to global public health. Therefore, new antimicrobial strategies are urgently needed to combat these resistant *K. pneumoniae*-related infections. Drug repurposing and combination are two effective strategies to solve this problem. By a high-throughput screening assay of FDA-approved drugs, we found that the potential small molecule 9-aminoacridine (9-AA) could be used as an antimicrobial alone or synergistically with rifampin (RIF) against extensively/pan-drug-resistant *K. pneumoniae*. In addition, 9-AA could overcome the shortcomings of RIF by reducing the occurrence of resistance. Mechanistic studies revealed that 9-AA interacted with bacterial DNA and disrupted the proton motive force in *K. pneumoniae*. Through liposomeization and combination with RIF, the cytotoxicity of 9-AA was significantly reduced without affecting its antimicrobial activity. In addition, we demonstrated the *in vivo* antimicrobial activity of 9-AA combined with RIF without detectable toxicity. In summary, 9-AA has the potential to be an antimicrobial agent or a RIF adjuvant for the treatment of multidrug-resistant *K. pneumoniae* infections.

**IMPORTANCE** *Klebsiella pneumoniae* is a leading cause of clinically acquired infections. The increasing occurrence of drug-resistant *K. pneumoniae* has posed a serious threat to global public health. We found that the potential small molecule 9-AA could be used as an antimicrobial alone or synergistically with RIF against drug-resistant *K. pneumoniae in vitro* and with low resistance occurrence. The combination of 9-AA or 9-AA liposomes with RIF possesses effective antimicrobial activity *in vivo* without detected toxicity. 9-AA exerted its antimicrobial activity by interacting with specific bacterial DNA and disrupting the proton motive force in *K. pneumoniae*. In summary, we found that 9-AA has the potential to be developed as a new antibacterial agent and adjuvant for RIF. Therefore, our study can reduce the risk of antimicrobial resistance and provide an option for the exploitation of new clinical drugs and a theoretical basis for the research on a new antimicrobial agent.

**KEYWORDS** *K. pneumoniae*, drug repurposing, drug combination, 9-aminoacridine, rifampin

*K*lebsiella pneumoniae is a leading cause of clinically acquired infections, causing respiratory, urinary, and blood infections (1, 2). Over the last decade, the increasing occurrence of extremely drug-resistant (XDR) and pan-drug-resistant (PDR) *K. pneumoniae* has posed a serious threat to global public health (3, 4). Drug-resistant *K. pneumoniae* infections are often associated with high mortality rates (5, 6). Since antimicrobial therapeutic options for multidrug-resistant *K. pneumoniae*-mediated infections are limited, it is urgent to develop novel antimicrobials or therapeutic strategies.

Address correspondence to Yong Wu, wuyong_zn@csu.edu.cn.

The authors declare no conflict of interest.

Drug repurposing and combination are two attractive strategies to combat bacterial resistance. The advantages of these strategies include enhanced antibacterial efficacy, reduced therapeutic dosage of a single drug, fewer adverse effects, prevented occurrence of drug resistance, sustainable cost, and faster clinical validation (7, 8).

Rifampin (RIF) is known to target bacterial polymerase. Disruption of cell wall integrity could promote RIF invasion and exhibit antimicrobial effects against Gram-negative bacteria (9). Thus, based on what is described above, a library containing 2,049 FDA-approved compounds was tested in combination with RIF by high-throughput assay to find drugs targeting cell wall permeability in our study. Through a series of phenotypic and mechanistic experiments, we repurposed a potential small molecule, 9-aminoacridine (9-AA), as an antimicrobial alone or synergistically with RIF against multidrug-resistant *K. pneumoniae*. Meanwhile, 9-AA exerted antibacterial effects by the specific bacterial DNA interaction and proton motive force (PMF) disruption. At present, 9-AA and its derivatives have also been reported or are in clinical trials for their efficacy in cancers, prion, and Alzheimer's disease (10–12). RIF is an efficient antibiotic against most Gram-positive bacteria and is a key drug used for the treatment of *Mycobacterium*. Clinically, RIF is not generally used for treating *K. pneumoniae* infection. In this study, 9-AA was found to synergize with RIF against *K. pneumoniae in vitro* and *in vivo* and decreased the resistance-inducing ability of RIF. By mechanistic study, we revealed the specific bacterial DNA interaction activities and PMF disruption by 9-AA. To the best of our knowledge, this is the first study to evaluate the antibacterial activity and its underlying mechanisms of 9-AA alone or combined with RIF against multidrug-resistant *K. pneumoniae*.

## RESULTS

**Antimicrobial molecule 9-AA against *K. pneumoniae* by high-throughput screening assay.** High-throughput screening was conducted as described above, and the flowchart is shown in Fig. 1A. After screening step 1, 111 hits were selected for their potential synergistic activity with RIF against *K. pneumoniae*. After excluding antibiotics and well-studied compounds among 111 hits, 16 hits were further subjected to the checkboard assay with RIF (supplemental file 2). Among these hits, 9-AA had a strong synergistic effect with RIF for all tested sensitive and multidrug-resistant *K. pneumoniae*, with fractional inhibitory concentration index (FICI) values of 0.375, 0.5, and 0.313, respectively (Fig. 1B). To further confirm the synergistic effects observed in the checkerboard assays, time-kill curves were measured (Fig. 1C). Monotherapy with the sub-MIC of 9-AA or RIF did not completely inhibit bacterial growth. In contrast, the 9-AA-RIF combination significantly inhibited the growth of ATCC 700603 and showed strong bactericidal effects against the XDR and PDR strains. Next, primary antimicrobial susceptibility tests of 9-AA against *Enterococcus faecium*, *Staphylococcus aureus*, *K. pneumoniae*, *Acinetobacter baumannii*, *Pseudomonas aeruginosa*, and *Enterobacter* species ("ESKAPE") pathogens are shown in Table 1. 9-AA had a strong antimicrobial effect on *K. pneumoniae*, with MICs from 8 to 16 $\mu$g/mL and minimal bactericidal concentrations (MBCs) from 16 to 64 $\mu$g/mL. However, 9-AA only exhibited moderate or no antimicrobial activity against other pathogens (Table 1). 9-AA was also found to enhance the antimicrobial activity of RIF against other type strains or clinical isolates of *K. pneumoniae*, with the MIC values of RIF decreased by 2- to 4-fold in the presence of sub-MIC (2 $\mu$g/mL) 9-AA (Table 2).

**Decreased antibiotic resistance-inducing ability of RIF by 9-AA.** It has been reported that RIF easily induces bacterial resistance (13, 14). The emergence of resistant mutants has also been reported when RIF was used in combination with other antibiotics (15, 16). To examine the effects of 9-AA on the resistance-inducing ability of RIF, multistep resistance selection and single-step resistance selection were applied. For multistep resistance selection, as shown in Fig. 2A, when *K. pneumoniae* ATCC 700603, KPWANG, and LH2020 were cultured in the presence of a sub-MIC of RIF for 25 passages, the MIC values were increased 16-, 128-, and 32-fold, respectively. However, those strains retained relatively stable MICs to RIF in the presence of the sub-MIC of 9-AA.

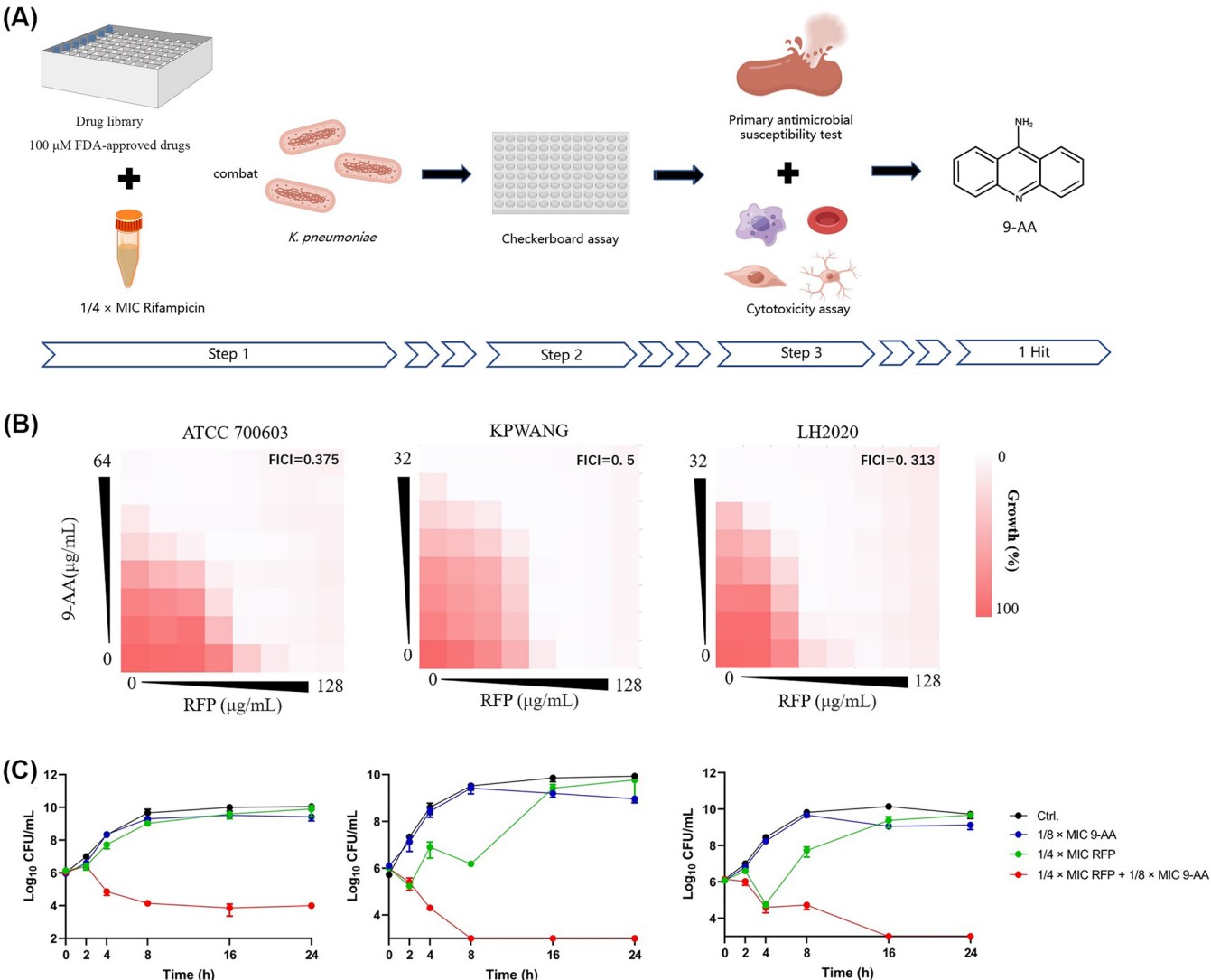

**FIG 1** 9-AA potentiated the antimicrobial activity of RIF against *K. pneumoniae*. (A) Flowchart of high-throughput screening. (B) Synergistic antimicrobial effects of 9-AA and RIF against *K. pneumoniae* ATCC 700603, XDR *K. pneumoniae* KPWANG, and PDR *K. pneumoniae* LH2020. All the checkerboard assays were repeated independently three times, and representative experiments are shown. (C) Killing kinetics of 1/8 × MIC 9-AA and 1/4 × MIC RIF alone or in combination against *K. pneumoniae* ATCC 700603, XDR *K. pneumoniae* KPWANG, and PDR *K. pneumoniae* LH2020.

Meanwhile, the MICs of 9-AA against the last-passage RIF-induced resistant *K. pneumoniae* strains remained unchanged or only slightly increased (Fig. 2B). Similarly, as shown in Fig. 2C, the combination of RIF and 9-AA remained highly synergistically sensitive against the RIF-induced resistant *K. pneumoniae* strains, with no elevated FICI observed. In accordance, for single-step resistance selection, the resistance induction rate of RIF in the sub-MIC 9-AA group was lower than that in the single RIF group (Fig. 2D and E). In general, RIF combined with the sub-MIC of 9-AA can overcome the shortcoming that RIF readily induces bacterial resistance.

**Bacterial DNA is a potential target of 9-AA.** We initially speculated that 9-AA enhanced RIF antibacterial activity by interacting with bacterial cell wall components. Thus, a peptide peptidoglycan (PGN) competitive inhibition assay was performed. However, as shown in Fig. S1A and Table S2 in the supplemental material, the MICs and MBCs, as well as the time growth curves, showed no difference between groups in the presence or absence of PGN. Similarly, no membrane-disrupting activity was observed in the presence of different concentrations of 9-AA through staining with the fluorescence probe SYTOX green (Fig. S1B). For bacterial outer membrane detection, as shown in Fig. S1C, the synergistic potentiation of 9-AA to RIF showed no change with

**TABLE 1** Antibacterial susceptibility of 9-AA against ESKAPE[a]

| Strain | 9-AA | | Resistance pattern | | | | | |
| --- | --- | --- | --- | --- | --- | --- | --- | --- |
| | MIC ($\mu$g/mL) | MBC ($\mu$g/mL) | Non-MDR | MDR | XDR | PDR | MSSA | MRSA |
| *K. pneumoniae* | | | | | | | | |
| ATCC 700603 | 16 | 64 | ✓ | | | | | |
| ATCC 4352 | 8 | 64 | ✓ | | | | | |
| ATCC 10031 | 8 | 64 | ✓ | | | | | |
| KPWANG | 16 | 16 | | | ✓ | | | |
| KPLUO | 16 | 16 | | | ✓ | | | |
| LH2020 | 16 | 16 | | | | ✓ | | |
| *P. aeruginosa* | | | | | | | | |
| PAO1 (ATCC 15692) | 128 | 128 | ✓ | | | | | |
| PA2530 | 64 | 128 | | | ✓ | | | |
| PA6930 | 64 | 128 | | | ✓ | | | |
| *A. baumannii* | | | | | | | | |
| ATCC 19606 | 64 | 128 | ✓ | | | | | |
| AB1095 | 16 | 32 | | | ✓ | | | |
| *E. coli* | | | | | | | | |
| ATCC 25922 | 8 | 16 | ✓ | | | | | |
| Y0064 | 32 | 64 | | | ✓ | | | |
| Y9592 | 16 | 32 | | | ✓ | | | |
| *S. aureus* | | | | | | | | |
| ATCC 43300 | 32 | 64 | | | | | | ✓ |
| USA 300 | 32 | 128 | | | | | | ✓ |
| ATCC 29213 | 32 | >128 | | | | | ✓ | |
| *S. epidermidis* | | | | | | | | |
| RP62A (ATCC 35984) | 32 | 64 | ✓ | | | | | |
| ATCC 12228 | 4 | 8 | ✓ | | | | | |
| *E. faecalis* | | | | | | | | |
| ATCC 29212 | 64 | 128 | ✓ | | | | | |
| *E. faecium* | | | | | | | | |
| U101 | 16 | 64 | ✓ | | | | | |

[a]MDR, multidrug resistant; XDR, extensively drug resistant; PDR, pan-drug resistant; MSSA, methicillin-susceptible *S. aureus*; MRSA, methicillin-resistant *S. aureus*.

the addition of the sub-MIC of $Mg^{2+}$ (an outer membrane strengthener) and decreased with the addition of the sub-MIC of EDTA (an outer membrane-abolishing reagent). The above-described results indicate that 9-AA does not target the cell wall component of *K. pneumoniae*.

We next examined the effect of 9-AA on bacterial DNA. Because of the fluorescence characteristics of 9-AA (Fig. S2), we speculated directly observing the 9-AA location in bacterial cells by confocal laser scanning microscopy (CLSM). Thus, the bacterial or mammalian cells were treated with 9-AA and then costained with FM4-64 (a membrane-staining dye)

**TABLE 2** 9-AA enhances the antimicrobial activity of RIF against *K. pneumoniae*

| Strain | Resistance pattern[b] | MIC ($\mu$g/mL) of: | | Fold change[a] |
| --- | --- | --- | --- | --- |
| | | RIF | RIF + 9-AA (2 $\mu$g/mL) | |
| ATCC 700603 | Non-MDR | 64 | 16 | 4 |
| ATCC 4352 | Non-MDR | 16 | 4 | 4 |
| ATCC 10031 | Non-MDR | 8 | 4 | 2 |
| KPWANG | XDR | 32 | 16 | 2 |
| KPLUO | XDR | 32 | 8 | 4 |
| LH2020 | PDR | 32 | 8 | 4 |

[a]Fold change of MIC = $MIC_{(RIF)}/MIC_{(RIF+9-AA)}$.
[b]MDR, multidrug resistant; XDR, extensively drug resistant; PDR, pan-drug resistant.

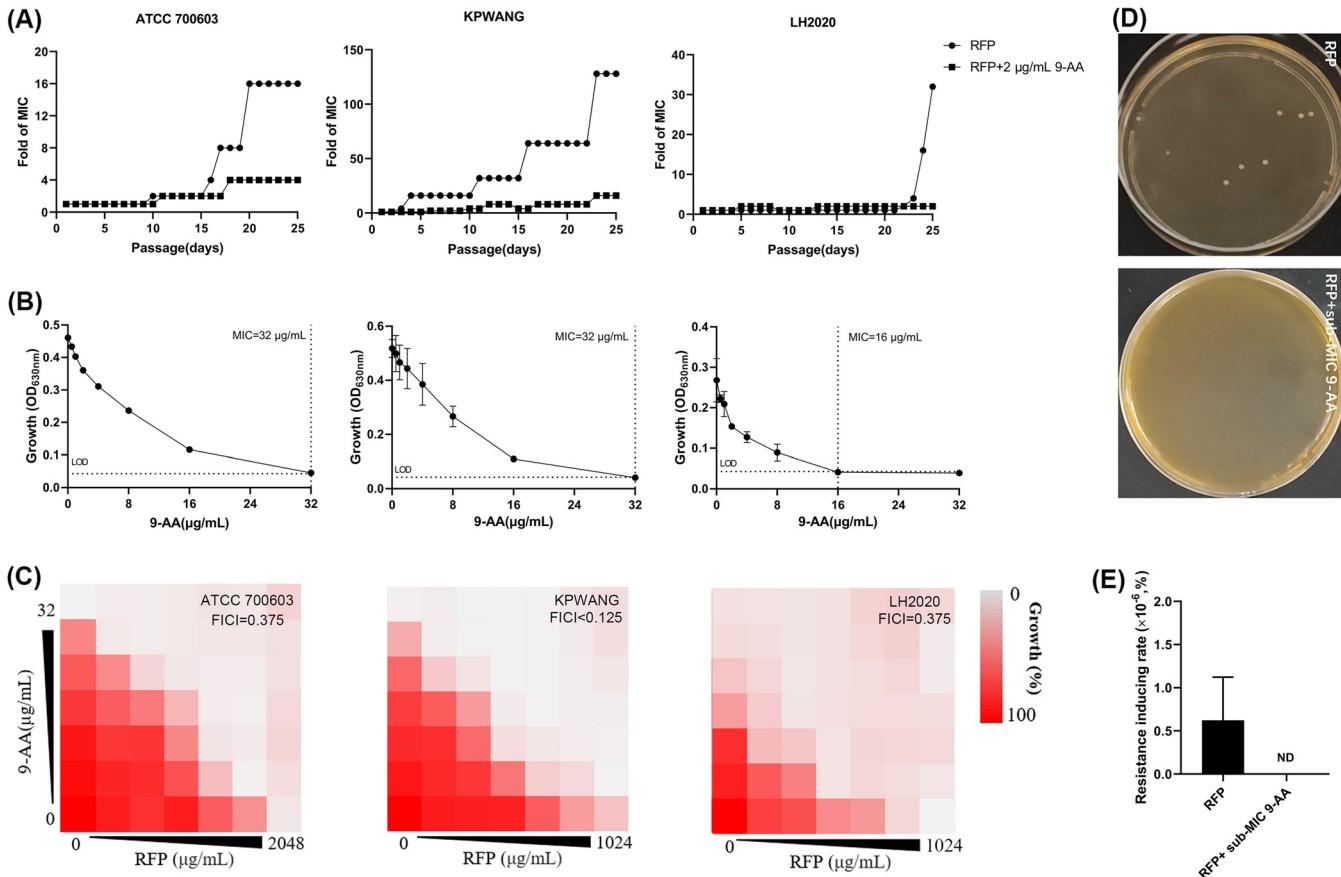

**FIG 2** 9-AA decreased the antibiotic resistance-inducing activity of RIF. (A) Consecutive resistance selections of *K. pneumoniae* ATCC 700603, XDR *K. pneumoniae* KPWANG, and PDR *K. pneumoniae* LH2020 by RIF or in the presence of 2 μg/mL 9-AA. (B) Growth inhibition effects of 9-AA against RIF-induced resistant strains *K. pneumoniae* after 25 days of passage. (C) Representative checkerboard images of 9-AA combined with RIF against the RIF-induced resistant strains. The checkerboard experiments were repeated thrice independently. (D) *K. pneumoniae* ATCC 700603 was subjected to single-step resistance selection in MH agar plates containing 64 μg/mL RIF alone or combined with 4 μg/mL 9-AA. (E) Quantification of the resistance induction rate. ND, not detected.

and propidium iodide (PI; a nuclear dye). CLSM images showed that the staining of 9-AA colocalized with both FM4-64 and PI in *K. pneumoniae* cells, which indicates that 9-AA may target both the cell membrane and DNA (Fig. 3A). However, as we mentioned above, no membrane disruption activity was observed by 9-AA through SYTOX green staining. Thus, we speculate that there was only an accumulation of 9-AA in the cell membrane, which probably further caused PMF disruption between the inner and outer membranes. As a control, the tetracycline (TET)-treated bacterial cells only exhibited colocalization with PI. In addition, as shown in Fig. 3A, no obvious infiltration was observed in mammalian cells by 9-AA. Transmission electron microscopy (TEM) images of 9-AA-treated bacteria showed that the overall bacterial morphology had obvious distortion and deformation, while dense material aggregated in the bacteria, which suggests possible chromatin aggregation (Fig. 3B). Similarly, the fluorescence measurement indicated that 9-AA led to a dose-dependent displacement of the DNA-binding probe SYTO 9 (Fig. 3C). By the addition of exogenous DNA, we found that the antimicrobial activity of 9-AA was significantly suppressed. The bacterial growth turbidity of the 9-AA plus 0.2 μg/mL DNA-treated group was significantly higher than that of the 9-AA group (Fig. 3D). Similar to the CFU counting, the total bacterial count of the 9-AA plus 0.2 μg/mL DNA-treated group was higher than that of the 9-AA-treated group (Fig. 3E). Then, molecular docking was carried out to simulate the binding mode between 9-AA and DNA. As shown in Table 3 and Fig. S3, 9-AA showed the best binding affinity to the DNA site of PDB accession no. 4U8A, with a predicted binding energy of −4.98 kcal/mol among these tested sites. The binding mode of 9-AA with PDB accession no. 4U8A is illustrated in Fig. 3F and G. In brief, 9-AA could

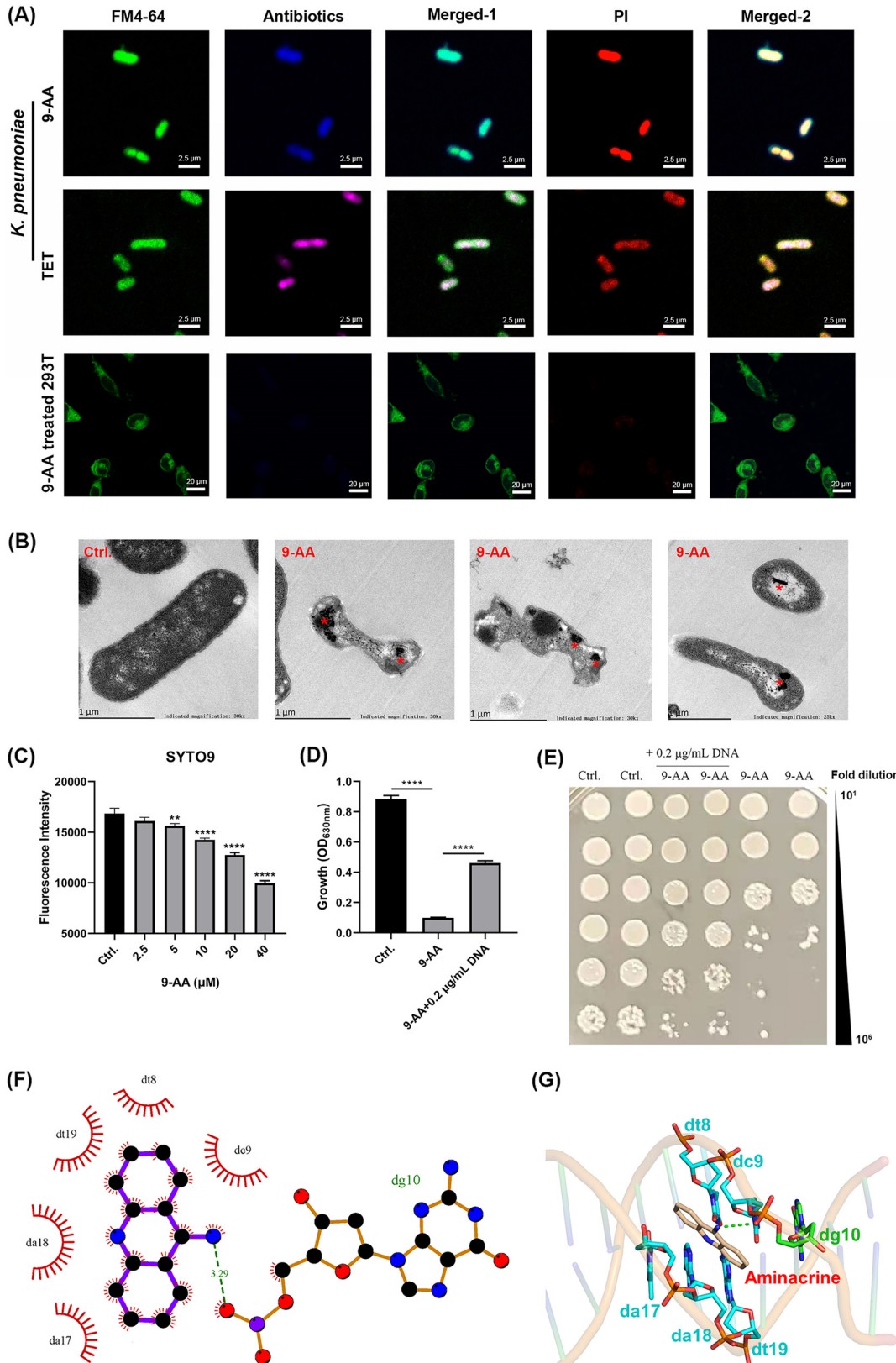

**FIG 3** 9-AA specifically targeted bacterial DNA. (A) Overlays of confocal microscopic images of *K. pneumoniae* ATCC 700603 and 293T cells stained by PI, FM4-64, and antibiotics. Fluorescence microscopic images in the first row, 32 $\mu$g/mL 9-AA treated *K. pneumoniae* ATCC 700603 costained with membrane dye FM4-64 (green), nuclear dye PI (red), and antibiotic 9-AA

**TABLE 3** Parameters of molecular docking

| Receptor | Binding energy (kcal/mol) | Intermolecular energy (kcal/mol) | Electrostatic energy (kcal/mol) | Internal energy (kcal/mol) | Predicted $K_i$ |
|---|---|---|---|---|---|
| 4U8A | −4.98 | −5.27 | −0.33 | +0.36 | 225.51 $\mu$M |
| T7 | −3.93 | −4.23 | −0.34 | +0.36 | 1.32 mM |
| CG | −3.30 | −3.60 | −0.19 | +0.36 | 3.80 mM |
| G10 | −4.01 | −4.30 | −0.44 | +0.36 | 1.16 mM |

interact with dg465 through a hydrogen bonding interaction and with dt81, da82, da83, da464, dg466, and dg477 through hydrophobic interactions. The molecular docking results further confirmed the stable binding of 9-AA to DNA. These results indicated that bacterial DNA is a potential target of 9-AA.

**PMF disruption by 9-AA.** By scanning electron microscopy (SEM) observation, thorn-like protrusions and depressions could be observed on the bacterial surface after treatment with 8× MIC of 9-AA (Fig. 4A), and the average cell diameter was also decreased after treatment (Fig. 4B). Considering that no membrane disruption activity was observed by 9-AA through SYTOX green staining, we speculate that there was an accumulation of 9-AA in the cell membrane, and it probably further caused PMF disruption between the inner and outer membranes. In bacteria, PMF is an electrochemical gradient of protons across the cell membrane and is indispensable to bacterial cellular processes such as ATP synthesis and transport of various solutes (17). PMF consists of the sum of two parts, the electric potential ($\Delta\Psi$) and the transmembrane proton gradient ($\Delta$ pH). The pH-sensitive probe BCECF-AM was used to measure $\Delta$pH, and 9-AA caused decreased fluorescence in a dose-dependent manner in *K. pneumoniae* (Fig. 4C), which indicated that the $\Delta$pH value increased. The membrane potential-sensitive dye DiSC$_3$(5) was used to measure $\Delta\Psi$, and 9-AA led to an increase in fluorescence (Fig. 4D), which indicated that $\Delta\Psi$ was dissipated. Because $\Delta\Psi$ and $\Delta$pH are interdependent, when one of the components is destroyed, the other will increase to keep the PMF constant (18). To further confirm the interaction with PMF by 9-AA, media with different pH values were prepared. As we expected, the increase in the external pH value enhanced the antibacterial effect of 9-AA (Fig. 4E and F). As previously reported, membrane depolarization and $\Delta\Psi$ disruption are related to the production of reactive oxygen species (ROS) (19). Similarly, in our study, 9-AA also promoted the accumulation of intracellular ROS (Fig. 4G), but the intracellular ATP levels significantly increased after treatment with 9-AA (Fig. 4H). Furthermore, as we expected, 9-AA showed partial synergy with TET, whose uptake is driven by $\Delta$pH (20, 21), and antagonized FeCl$_3$, which increased $\Delta\Psi$ (Fig. 4I). In addition, as shown in Fig. 4J, 9-AA showed significant surface motility inhibition activity against *K. pneumoniae*. As previously reported, surface motility was powered by the PMF (22, 23).

**Acceptable *in vitro* toxicity profile of 9-AA.** The cytotoxicity of 9-AA was assessed by hemolysis assays and a Cell Counting Kit-8 (CCK-8) (Dojindo, Japan) kit. 9-AA did not exhibit hemolytic activity on red blood cells (RBCs) even at concentrations up to 64 $\mu$g/mL (Fig. 5A). As observed by phase-contrast microscopy, the morphology of

**FIG 3** Legend (Continued)
(dark blue). Fluorescence microscopic images in the second row, 32 $\mu$g/mL TET-treated *K. pneumoniae* ATCC 700603 costained with membrane dye FM4-64 (green), nuclear dye PI (red), and antibiotic TET (purple). Fluorescence microscopic images in the third row, 32 $\mu$g/mL 9-AA-treated 293T cells costained with membrane dye FM4-64 (green), DNA dye PI (red), and antibiotic 9-AA (dark blue). Merged-1 column images represent the merge between the first and second columns. Merged-2 column images represent the merge between the third and fourth columns. The scale bar in the bacterial images marks 2.5 $\mu$m, and the scale bar in the cell images marks 20 $\mu$m. All fluorescence images are shown in pseudocolor to better distinguish different fluorescence. (B) Representative TEM images of *K. pneumoniae* ATCC 700603 treated with 8× MIC 9-AA for 1 h. Red asterisks indicate dense material aggregated in the bacteria. Scale bar, 1 $\mu$m. (C) Fluorophotometric measurement showing that 9-AA was able to competitively displace 5 $\mu$M of the DNA-binding probe SYTO 9 in a dose-dependent manner. (D, E) Exogenous DNA (0.2 $\mu$g/mL) inhibited the antimicrobial effect of 1× MIC 9-AA against *K. pneumoniae* ATCC 700603, as shown by the OD$_{630}$ and viable cell counts. (F) Two-dimensional schematic of 9-AA binding to DNA (PDB accession no. 4U8A). Green imaginary lines indicate hydrogen bonds, and red gears indicate hydrophobic interactions. (G) Three-dimensional schematic of 9-AA binding to DNA (PDB accession no. 4U8A), and the green dotted lines represent the hydrogen bond. **, $P < 0.01$; ****, $P < 0.0001$.

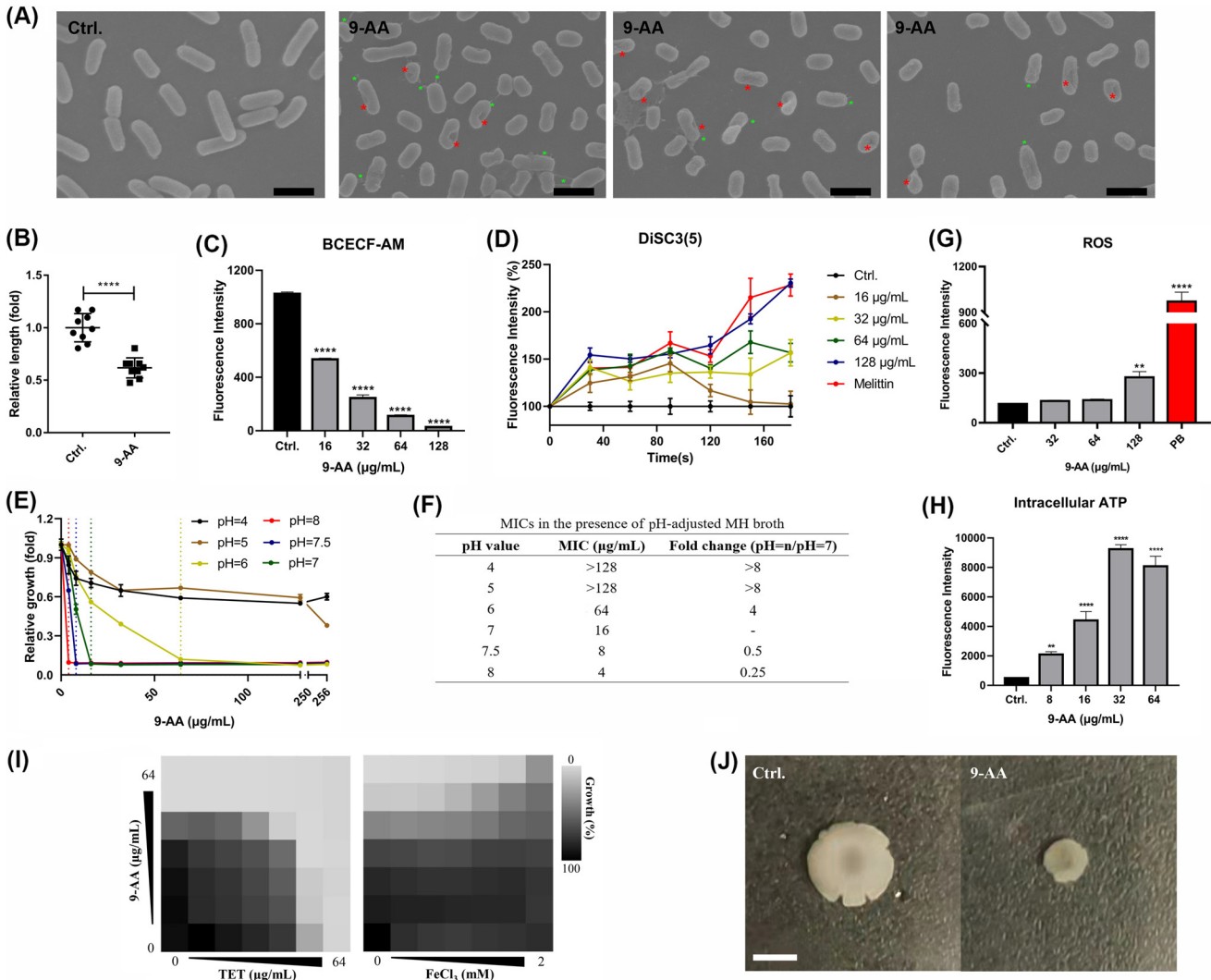

**FIG 4** 9-AA disrupted PMF by affecting the bacterial electric potential. (A) SEM images of *K. pneumoniae* treated with 8× MIC og 9-AA for 1 h. Green and red asterisks indicate thornlike protrusions and depressions on the bacterial surface, respectively. Scale bar, 2 $\mu$m. (B) Quantification of relative bacterial length in the SEM. (C) The intracellular pH of *K. pneumoniae* was determined using a BCECF-AM fluorescent probe. (D) The electric potential of *K. pneumoniae* was determined using a DiSC$_3$(5) fluorescent probe. (E) Enhanced bacterial growth inhibition effect of 9-AA in medium with increased pH. Relative growth (fold) is calculated as follows: OD$_{630}$ of the 9-AA-treated group/mean OD$_{630}$ of the control group at the same pH. The concentration of 9-AA at the intersection of different-colored dotted lines and abscissa is the MIC of 9-AA to *K. pneumoniae* at the corresponding pH. (F) Effect of external pH alteration on the MIC values of 9-AA. (G) Accumulation of intracellular ROS after treatment with 9-AA. (H) Increased levels of intracellular ATP in *K. pneumoniae* after treatment with 9-AA. (I) Checkerboard graphs of 9-AA combined with TET and FeCl$_3$. The checkerboard experiments were repeated independently three times, and representative experiments are shown. (J) Surface motility of *K. pneumoniae* in the absence or presence of 9-AA. **, $P < 0.01$; ****, $P < 0.0001$.

128 $\mu$g/mL 9-AA-treated RBCs was normal compared to that of the control group, but obvious shrinkage was observed after Triton X-100 (positive control) treatment (Fig. 5B). To improve the safety of 9-AA, we constructed 9-AA(L). The characterization of 9-AA(L) was well studied and shown in Materials and Methods and supplemental file 1. The cell viability of human normal and tumor cell lines after treatment with 9-AA or 9-AA/L is shown in Fig. 5C, and the related 50% inhibitory concentrations (IC$_{50}$s) are listed in Table S3. 9-AA showed moderate cytotoxicity to mammalian cells, but when used as a RIF adjuvant, the reduced dosage of 9-AA further diminished its toxicity. Liposome-encapsulated drugs in lipid bilayers are one of the delivery systems that can effectively reduce drug toxicity (24, 25). As we expected, 9-AA(L) significantly relieved its cytotoxicity and increased its IC$_{50}$ values (Fig. 5C and D; Table S3). Collectively, 9-AA has acceptable toxicity as an adjuvant or 9-AA(L).

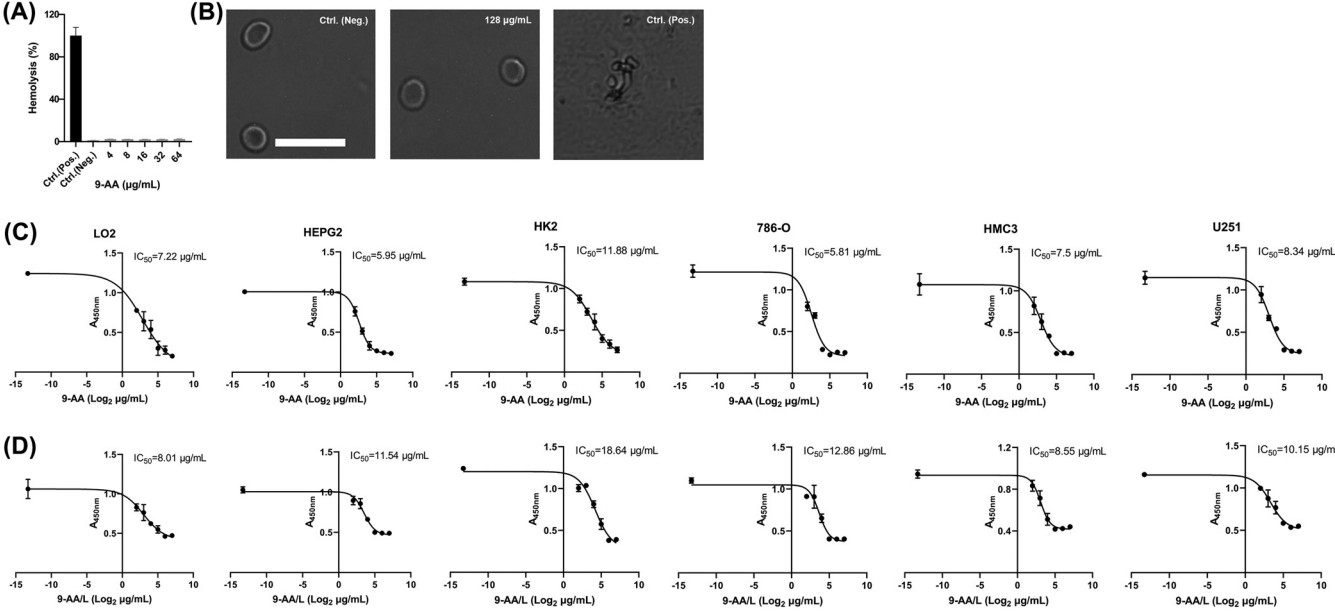

**FIG 5** Cytotoxicity profile of 9-AA. (A) RBCs were treated with the indicated concentrations of 9-AA (0 to 64 $\mu$g/mL). DMSO (0.5%) and Triton X-100 (0.1%) were used as negative and positive controls, respectively. (B) Representative images of RBCs treated with 128 $\mu$g/mL 9-AA, 0.5% DMSO, or 0.1% Triton X-100. (C, D) Cell viability of normal (LO2, HK2, HMC3) and tumor (HepG2, 786-O, U251) human cell lines after treatment with 9-AA or 9-AA(L). IC$_{50}$, related 50% inhibitory concentration.

***In vivo* antimicrobial activity of 9-AA and 9-AA(L).** We constructed the wound infection model and skin subcutaneous abscess model following the flowchart in Fig. S5. Then, the antimicrobial efficacy of 9-AA alone or in combination with RIF was assessed by viable bacterial cell counting and hematoxylin and eosin (H&E) staining. In the wound infection model, the counting of the 0.5% (wt/wt) 9-AA- or RIF ointment-treated group showed no or moderate CFU decrease, but the combination group significantly reduced the bacterial counts by 3.18 log$_{10}$ (versus vehicle) and 1.12 log$_{10}$ CFU/mL (versus the RIF-treated group), respectively (Fig. 6A). In subcutaneous skin abscesses, a single dose of 15 mg/kg 9-AA decreased the abscess area compared with that in the vehicle group (Fig. 6B). By CFU counting, a single dose of 15 mg/kg 9-AA or 20 mg/kg RIF did not significantly reduce the viable bacterial loads in the abscess; however, the combined use of the drugs significantly decreased the bacterial load by 3.15 log$_{10}$ CFU/abscess (Fig. 6C). Consistently, H&E staining images showed that significantly reduced inflammatory infiltration was observed in the combination group compared to the monotherapy group (Fig. 6D).

Furthermore, the antibacterial activity of 9-AA liposomes *in vivo* was assessed. After treatment with 15 mg/kg 9-AA(L), 7.5 mg/kg plus 7.5 mg/kg 9-AA plus RIF(L), and 15 mg/kg plus 15 mg/kg 9-AA plus RIF(L), the bacterial loads in the abscess were reduced by 0.80, 1.62, and 2.90 log$_{10}$ CFU/abscess, respectively (Fig. 6E). Consistently, H&E-stained sections of skin tissues from liposome-treated mice revealed no inflammatory infiltration (Fig. 6F).

Meanwhile, the *in vivo* toxicity was also determined. No statistical significance was detected for hematological liver functional, renal functional, or myocardial functional biomarkers between the drug-treated and vehicle groups (Fig. S6A and B). Similarly, the H&E staining of these organs with 9-AA and 9-AA(L) treatment showed no pathological changes (Fig. S6C).

**Synergistic antibacterial effects between 9-AA analogs and RIF.** As shown in Fig. 7A, the antimicrobial synergistic effects against *K. pneumoniae* were also observed in 9-AA analogs in combination with RIF. Different from the combination of 9-AA with RIF, although the combinations of 9-AA analogs and RIF were efficient in bacterial growth inhibition, all these combinations lacked bactericidal activity against *K. pneumoniae* during

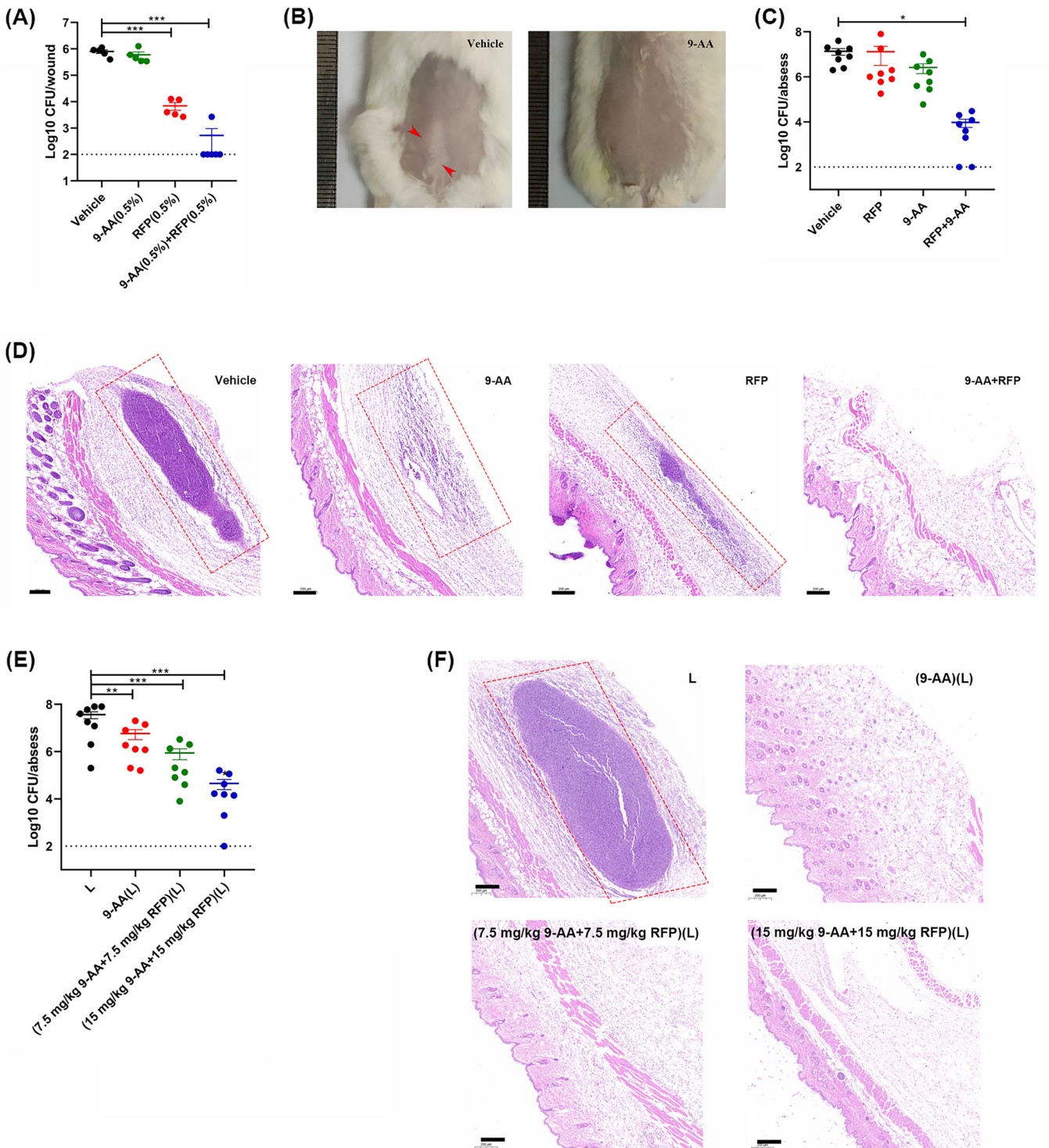

**FIG 6** *In vivo* antimicrobial activity of 9-AA and 9-AA(L). (A) Bacterial load quantification after treatment with ointment containing 9-AA and RIF in the acute wound infection model. Data are expressed as the means ± SDs; *n* = 5 to 6/group. (B) Representative abscess images of the vehicle-treated group and 15 mg/kg 9-AA-treated group. Red arrows indicate skin abscesses. (C, D) Bacterial counts (C) and representative H&E staining images (D) of the abscesses after treatment with 9-AA (15 mg/kg) and RIF (20 mg/kg) alone or in combination. Scale bar, 200 $\mu$m. Data are expressed as the means ± SDs; *n* = 8/group. (E, F) Bacterial counts (E) and H&E staining (F) of the abscesses after treatment with 15 mg/kg 9-AA(L), 7.5 mg/kg 9-AA plus 7.5 mg/kg RIF(L), or 15 mg/kg 9-AA plus 15 mg/kg RIF(L). Scale bar, 200 $\mu$m. Data are expressed as the means ± SDs; *n* = 8/group. *, *P* < 0.05; **, *P* < 0.01; ***, *P* < 0.001.

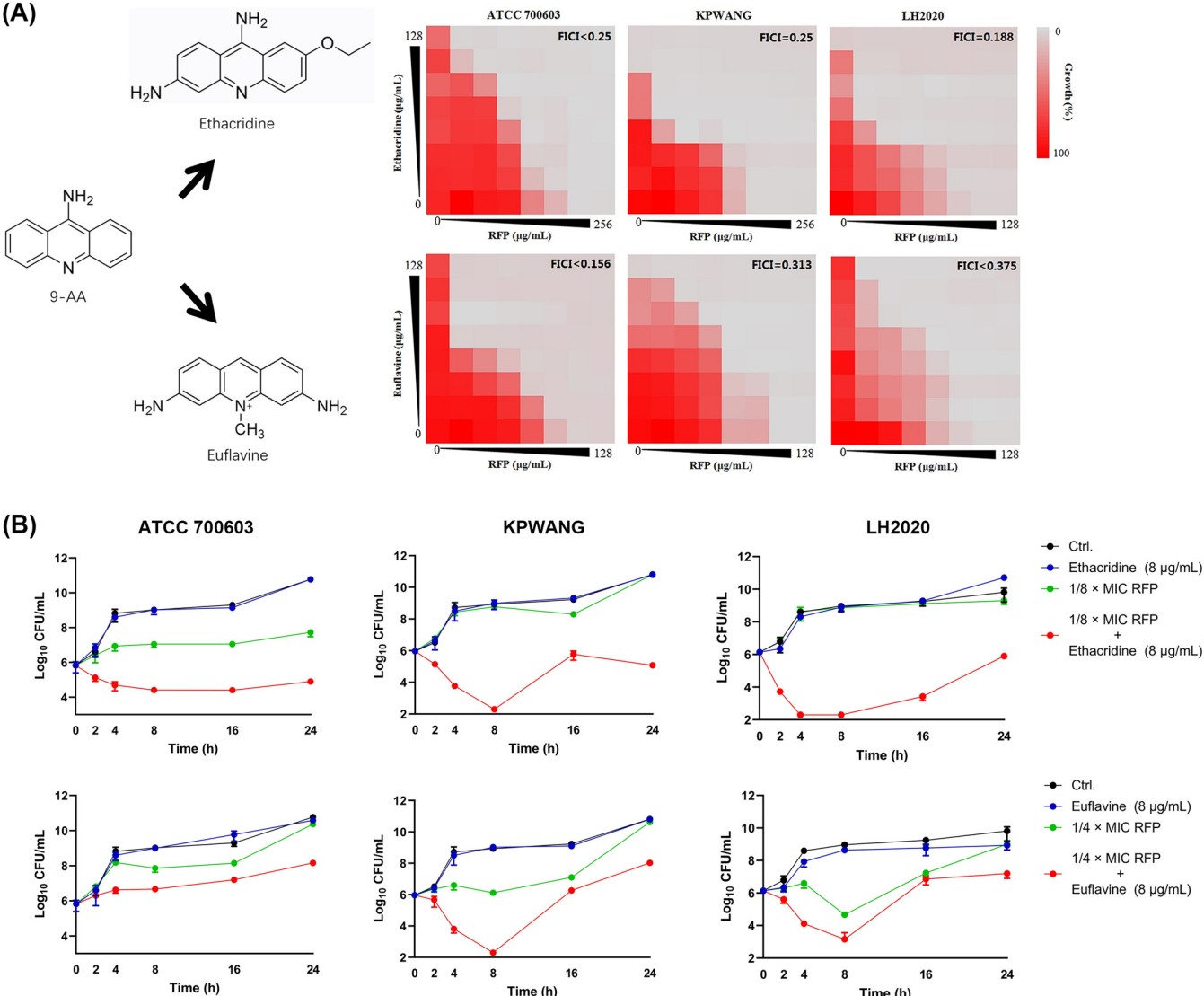

**FIG 7** Synergistic antibacterial effects between 9-AA analogs and RIF against *K. pneumoniae*. (A) Checkerboard graphs of 9-AA analogs and RIF against *K. pneumoniae* ATCC 700603, XDR *K. pneumoniae* KPWANG, and PDR *K. pneumoniae* LH2020. All checkerboard assays were repeated independently three times. (B) Killing kinetics of sub-MIC 9-AA analogs and sub-MIC RIF alone or in combination against *K. pneumoniae* ATCC 700603, XDR *K. pneumoniae* KPWANG, and PDR *K. pneumoniae* LH2020.

the 24-h treatment period (Fig. 7B). In addition, the cytotoxicity of 9-AA analogs was measured in various cell lines. However, the analogs were more toxic to most of the tested human cell lines than 9-AA (Table S3). Further, the checkerboard assay was performed to assess the combinational antimicrobial effects of 9-AA and 9-AA analogs with other conventional antibiotics against *K. pneumoniae*. As shown in Fig. S7, no synergistic effect was found between the analogs and conventional antibiotics except for the combination of ethacridine and doxycycline. These results suggested that the synergistic activity between 9-AA and 9-AA analogs with RIF could be antibiotic specific. In summary, 9-AA has the greatest potential to develop as an antibacterial agent among its analogs.

## DISCUSSION

*K. pneumoniae* causes a wide range of infections, and the rapid spread of multidrug-resistant *K. pneumoniae* strains has made treatment increasingly difficult. Our study repurposes the potential small molecule 9-AA alone or combined with RIF, which exhibits strong antimicrobial activity against extensively or pan-drug-resistant *K. pneumoniae*

through specific bacterial DNA interactions and PMF disruption. Meanwhile, 9-AA combined with RIF also displayed effective antimicrobial activity *in vivo* without detectable toxicity. To the best of our knowledge, this is the first study to systematically investigate the potential of 9-AA as an antibacterial agent or an antimicrobial adjuvant with RIF against multidrug-resistant *K. pneumoniae* strains in *vitro* and in *vivo*.

Recent studies on repurposing 9-AA and its derivatives have more often focused on their effectiveness in treating cancers, prions, and Alzheimer's disease (10–12). Some studies have also shown that 9-AA coupled to Arabic gum dialdehyde and 9-AA hydrochloride hydrate conjugated with nanoparticles enhanced their antimicrobial activities (26, 27); however, these studies on the antibacterial effect of 9-AA were limited to type strains and *in vitro* nonmechanism experiments.

9-AA enhanced the antibacterial activity of RIF against both sensitive and multidrug-resistant *K. pneumoniae*. Although RIF can easily induce resistant mutant strains (13, 14) and the emergence of resistant mutants when RIF was used in combination with other antibiotics has also been reported (15, 16), in our study, when combined with the sub-MIC of 9-AA, RIF had a relatively low tendency for resistance development. Though the MIC values of *K. pneumoniae* ATCC 700603 and KPWANG in the combinational group had some degree of increase by multistep resistance selection, there was still no drug resistance that emerged in short-term treatment (within 10 passages) (Fig. 2A). These results suggest that 9-AA suppressed the development of RIF resistance. Also, 9-AA and RIF combination still showed synergistic antibacterial activity against RIF-induced resistant *K. pneumoniae* strains (Fig. 2C), which indicated that the RIF resistance mutation did not easily affect the synergistic antibacterial activity of 9-AA and RIF.

Previous studies showed 9-AA is a mutagen which target nucleic acids and cause gene mutations (28–30). However, comparing nucleotide incorporation assay and gel electrophoresis used in previous research, CLSM and SEM images in our present study showed 9-AA targeted bacterial DNA more intuitively. Molecular docking can better predict the binding interaction and evaluate the interaction forces of 9-AA and bacterial DNA. Since RIF targets the DNA-dependent RNA polymerase to inhibit DNA transcription (14, 31), 9-AA and RIF may synergistically inhibit DNA function to exert antibacterial effect. In addition, we found that PMF was also a potential target of 9-AA. 9-AA dissipated $\Delta\Psi$, and then the $\Delta$ pH increased compensatively. When compounds disrupt $\Delta\Psi$, a synergism effect will be observed in combination with antibiotics which are driven by $\Delta$ pH, while antagonism will show in combination with antibiotics whose uptake depends on $\Delta\Psi$ (32, 33). This provides us with new strategies to find new antimicrobial combinations, for example, 9-AA and tetracycline antibiotics (Fig. 4I). PMF disruption was also reported to be associated with the recovery of antibiotic potentiation (34). Thus, 9-AA exerted its antibacterial effect through dual mechanisms, interacting with specific bacterial DNA and disrupting the PMF in *K. pneumoniae*. Similarly, several studies have previously reported dual-mechanism antibacterial agents. Polyhexamethylene biguanide functions by targeting the bacterial membrane and DNA (35). Alternatively, amphiphilic CoB oligoguanidine has a dual mechanism of action that involves membrane disruption and DNA inhibition (36). SCH-79797 and its derivative IRS-16 have two independent cellular targets, bacterial membrane and folate metabolism (37). Telavancin is a semisynthetic lipoglycopeptide that inhibits cell wall synthesis and disrupts bacterial cell membrane barrier functions (38). Antibiotics with dual mechanisms are potential agents to address drug resistance and have advantages, including broad-spectrum efficacy, excellent therapeutic index, low resistance generation rate, and excellent *in vivo* activity. In our study, 9-AA also showed these strengths. DNA-targeted drugs may induce DNA lesions in human normal cells and further lead to diseases such as cancer. But in our study, 9-AA is not easy to damage the DNA of normal cells. First, in the DNA mechanistic experiment, no obvious 9-AA infiltration was observed in mammalian cells during the assay, while the bacterial cell membrane and DNA were bound by 9-AA (Fig. 3A). It is indicated 9-AA could selectively target bacteria in a short time treatment. Second, our study focuses on the localized infection model and topical medication, it is not easy to produce systemic and long-term toxicity.

Third, drug combination can lower the drug dosage and reduce potential toxicity, including cytotoxicity and genotoxicity. Finally, the clinical course of one antibiotic is often shorter than 2 weeks; otherwise, it will cause flora imbalance and drug resistance (39). Also, a previous study has reported the low genotoxicity of 9-AA (40). We will continue to conduct further structural modifications of 9-AA to improve its efficacy and safety.

Skin and soft tissue infections (SSTIs) are significant clinical and economic burdens in health care facilities (41, 42). A SENTRY antimicrobial surveillance report (North America) noted that *Klebsiella* spp. make up 5.1% of the pathogens isolated from SSTIs (43). *K. pneumoniae* has been increasingly proposed as a cause of SSTIs, and multidrug-resistant strains make treatment more challenging. Meanwhile, few studies reported infections associated with high bacterial densities ($>10^7$ CFU/mL bacteria). Hence, we constructed a high-bacterial-density wound infection model and skin subcutaneous abscess model. As we expected, 9-AA and 9-AA(L) combined with RIF exhibited strong *in vivo* antibacterial activity.

Antibiotics entrapped in liposomes significantly enhance their antibacterial activity and pharmacokinetic properties and reduce toxic side effects during antibiotic therapy (44, 45). Tobramycin encapsulated in liposomes to treat *P. aeruginosa*-associated chronic pulmonary infection in mice was more effective due to improved pharmacokinetics, sustained release and accumulation of tobramycin at infection sites, and reduced nephrotoxicity (24). Liposomal amphotericin B, an antifungal agent, is the first liposomal drug launched on the market. Liposomal amphotericin B can reduce the risk of renal toxicity of drugs and can be safely used in antibacterial therapy (25). Consistently, 9-AA(L) improved the antimicrobial effect and reduced the toxicity of 9-AA in our study. The derivatives of 9-AA also showed synergistic antibacterial effects with RIF against sensitive and multidrug-resistant *K. pneumoniae*, which could also be valuable in future studies. In conclusion, 9-AA has the potential to be developed as a new antibacterial agent and adjuvant for RIF. However, before that, structural optimization of these compounds should be performed to improve their efficacy and biosafety.

## MATERIALS AND METHODS

**Bacterial isolates, media, and reagents.** The strains used in this study are listed in Table S1 in the supplemental material. Some of the clinical strains were collected from the Third Xiangya Hospital of Central South University (Changsha, China). The clinical isolates were identified using a Vitek-2 compact system (bioMérieux, France) and a matrix-assisted laser desorption ionization–time of flight (MALDI-TOF) mass spectrometer (Bruker Daltonics, Bremen, Germany). Unless otherwise stated, these strains were cultured in corresponding media, such as tryptic soy broth (TSB) for *Staphylococcus*, Luria-Bertani (LB) broth for Gram-negative strains, and brain heart infusion (BHI) for *Enterococcus*. These media were all purchased from Solarbio (Shanghai, China). Antimicrobials used in this study were obtained from MedChemExpress (NJ, USA) and dissolved in deionized water or dimethyl sulfoxide (DMSO) according to the instructions.

**High-throughput screening.** To obtain the potential antimicrobials, a drug library containing 2,049 FDA-approved compounds was selected for three rounds of screening tests. First, 99 $\mu$L of Mueller-Hinton (MH) broth containing $1 \times 10^5$ CFU/mL *K. pneumoniae* ATCC 700603 with 16 $\mu$g/mL RIF (1/4× MIC) was added to a 96-well plate (Corning Costar, USA), and 1 $\mu$L of library compounds was added to the above-described well plates to achieve a final concentration of 100 $\mu$M. After incubation at 37°C for 18 h, the bacterial viability was examined by optical density at 630 nm ($OD_{630}$) using a microplate reader (Bio-Rad iMark; USA). Compounds with growth inhibition activity greater than 80% compared with the control group were selected for the next screening. Next, a checkerboard assay was conducted, and compounds exhibiting synergy with RIF were selected for final screening. In the last step, the MICs and minimal bactericidal concentrations (MBCs) of the compounds against *K. pneumoniae* were determined. Meanwhile, the cytotoxicity of the compounds against normal human cells and tumor cells was also tested by Cell Counting Kit-8 (CCK-8) assay and hemolysis assay, respectively. Finally, 9-AA was selected for further study (Fig. 1A). The flow chart of the High-throughput screening was plotted using Figdraw.

**Antimicrobial susceptibility test.** The MICs and MBCs of 9-AA and RIF were determined by a standard broth microdilution assay according to the recommendations of the Clinical and Laboratory Standards Institute (46). Briefly, log-phase bacteria were diluted with MH broth to approximately $1 \times 10^6$ CFU/mL. Next, 50 $\mu$L of the bacterial suspension and 50 $\mu$L of 2-fold serially diluted antimicrobials were added to a 96-well cell plate. After incubation at 37°C for 16 to 18 h, the bacterial turbidity ($OD_{630}$) was examined using a microplate reader. The MIC was defined as the lowest concentration inhibiting measurable bacterial growth. Meanwhile, an aliquot of bacterial suspension from 1× MIC was spotted on sheep blood agar plates (AutoBio, Zhengzhou, China), and the MBC was defined as the lowest concentration at which no bacterial colonies grew on the plate after incubation at 37°C for 24 h.

**Checkerboard assay.** Drug combinations were assessed by checkerboard assays according to the CLSI guidelines (46). Briefly, 2-fold serial dilutions of the two compounds were prepared in MH broth and added to the columns and rows of a 96-well cell plate. Then, MH broth-diluted bacterial suspensions were added to each well to achieve a final cell density of $5 \times 10^5$ CFU/mL/well. The plates were incubated for 16 to 18 h at 37°C, and the MICs were measured at $OD_{630}$ on a microplate reader. The compound interactions were analyzed by the fractional inhibitory concentration index (FICI). The FICI was calculated according to the following equation:

$$\text{FICI} = \frac{\text{MIC}_A(\text{combination})}{\text{MIC}_A(\text{alone})} + \frac{\text{MIC}_B(\text{combination})}{\text{MIC}_B(\text{alone})}$$

MICs (combination) are the lowest MIC values when combined, and MICs (alone) are the concentrations of compounds A and B when used alone. The FICI was interpreted as follows: FICI $\leq 0.5$, synergy; $0.5 < \text{FICI} \leq 1$, partial synergy; $1 < \text{FICI} \leq 4$, indifference; and FICI $> 4.0$, antagonism (47). In addition, for some mechanism studies, the checkerboard experiments were added with 1.25 mM EDTA or 10 mM $Mg^{2+}$ before tests.

**Time-kill assay.** In a typical assay, bacteria (*K. pneumoniae* ATCC 700603 and the clinical isolates of KPWANG or LH2020) were cultured overnight in LB broth at 37°C with shaking at 180 rpm. The cells were subcultured to the log growth phase in the same broth. The bacterial concentration was adjusted to $1 \times 10^6$ CFU/mL in fresh MH broth and exposed to RIF and 9-AA alone or in combination. MH broth exposed to 0.1% DMSO served as a control. Viable counts were determined by agar plate counting at the time points of 0, 2, 4, 8, 16, and 24 h.

**Bacterial resistance detection.** Consecutive and single-step resistance selection was performed to evaluate the antimicrobial resistance-inducing ability of RIF alone or combined with 9-AA against *K. pneumoniae*. For consecutive resistance selection, briefly, the MIC values of RIF alone or combined with 2 μg/mL 9-AA against *K. pneumoniae* were determined using the antimicrobial susceptibility test as described above. Then, cultures from the $1/2\times$ MICs were diluted 1:1,000 into fresh MH broth. The diluted bacterial suspensions were continuously used for the second day, and the MICs were measured again and repeated for 25 days. At the end of the experiment, the obtained RIF-resistant strains of the last generation were used for antimicrobial susceptibility testing by 9-AA alone or combined with RIF (13). For single-step resistance selection, *K. pneumoniae* ATCC 700603 cultures grown overnight were adjusted to an $OD_{630}$ of 0.5. One hundred microliters of the bacterial suspension was spread evenly onto MH agar plates containing $1\times$ MIC RIF alone or combined with sub-MIC 9-AA. After incubation at 37°C for 48 h, the CFU on the plates were counted, and the frequency of resistance was calculated (48).

**Peptide peptidoglycan competitive inhibition assay.** The MICs and time growth curve of 9-AA in the presence of the purified cell wall component PGN were measured as described above in the "Antimicrobial susceptibility test" and "Time-kill assay" sections with minor modifications. Namely, before adding bacterial suspensions, the tested compounds and PGN were preincubated at 37°C for 1 h (49).

**Membrane permeability assay.** *K. pneumoniae* ATCC 700603 was grown to log phase, washed, and resuspended in a 0.5 McFarland standard in 5 mM HEPES buffer (pH 7.2). SYTOX green (Thermo Fisher Scientific, USA) was added to the suspensions to a final concentration of 5 μM. After 30 min of incubation in the dark, 90 μL of the suspension and 10 μL of 9-AA at the indicated concentrations were added to a black 96-well plate. The fluorescence intensity was measured as the time elapsed using a microplate reader (PerkinElmer EnVision; USA). The excitation/emission wavelengths were 485 nm/525 nm, respectively, for SYTOX green (50).

**Fluorescence characteristics of 9-AA.** The fluorescence emission spectrum of 9-AA was detected using a spectrofluorometer. The excitation wavelength for 9-AA was set as 390 nm. The 9-AA-labeled bacteria and their fluorescence were observed by confocal laser scanning microscopy (CLSM; Zeiss LSM800; Jena, Germany). In brief, bacterial cultures were washed, resuspended in phosphate-buffered saline (PBS), and treated with 32 μg/mL 9-AA for 30 min. Then, bacterial cells were transferred to microscope slides and observed by CLSM.

**CLSM observation of bacteria and mammalian cells stained by 9-AA.** Log-phase *K. pneumoniae* ATCC 700603 cells were harvested after centrifugation at 4,000 rpm for 8 min, washed three times with PBS, and resuspended in PBS to an $OD_{630}$ of 0.2. 9-AA or tetracycline (TET; positive control for DNA interaction) was added at a final concentration of 32 μg/mL. The bacterial suspensions were cultured at 37°C for 60 min, followed by centrifugation at 4,000 rpm for 8 min. The precipitates were washed and resuspended in PBS with 10 μg/mL propidium iodide (PI) and stained for 15 min at room temperature. Subsequently, the bacteria were washed and resuspended in PBS with FM4-64 dye (5 μg/mL) and stained on ice for 1 min. Finally, after removing excess dye by PBS washing, the bacteria were transferred to slides for confocal imaging (35). The fluorescence intensity was measured at excitation and emission wavelengths of 390 nm and 490 nm, 270 nm and 514 nm, 515 nm and 640 nm, and 535 nm and 617 nm for 9-AA, TET, FM4-64, and PI, respectively, using a microplate reader (PerkinElmer EnVision; USA).

293T cells were seeded into 6-well culture plates plated with cell-climbing slices and incubated at 37°C in 5% $CO_2$ for 6 h for attachment. Then, 32 μg/mL 9-AA was added to the culture and incubated for 60 min. The culture medium was removed, and the cell-climbing slices were washed with PBS 3 times. Then, cell-climbing slices were incubated with 10 μg/mL PI for 15 min. The medium was removed, and the cell-climbing slices were washed again with PBS. The slices were stained with FM4-64 at a final concentration of 5 μg/mL in PBS for 1 min. Subsequently, the slices were washed three times with PBS, affixed to microscope slides, and observed with CLSM.

**SYTO 9 displacement assay.** To investigate the binding target of 9-AA, the SYTO 9 displacement assay was performed as previously reported (35) with minor modifications. Briefly, mid-log-phase *K. pneumoniae* ATCC 700603 was washed three times and adjusted to $OD_{630}$ of 0.2 with HEPES. SYTO 9 (a nucleic acid-specific fluorescent dye) was added to the bacterial suspension to a final concentration of 5 $\mu$M and incubated at room temperature for 15 min in the dark. Then, different concentrations of 9-AA were added to the above-described bacterial suspension and coincubated for another 15 min. Meanwhile, the non-DNA-binding antibiotic gentamicin was used as the control. The fluorescence intensity was measured using a microplate reader with excitation and emission wavelengths of 485 and 498 nm, respectively.

**Exogenous DNA competition assay.** A single colony of *K. pneumoniae* ATCC 700603 was inoculated in fresh LB medium and cultured overnight. The bacterial cells were adjusted to a 0.5 McFarland standard turbidity and diluted 100-fold in fresh MH broth. Then, 16 $\mu$g/mL 9-AA in the presence or absence of 0.2 $\mu$g/mL DNA was added to the bacterial suspension. After 24 h of incubation at 37°C and 180 rpm, the samples were serially diluted and spotted on MH agar plates for overnight incubation, and the $OD_{630}$ values were also detected to evaluate the inhibitory effect of DNA on 9-AA antimicrobial activity.

**Molecular docking.** The molecular structure of 9-AA was obtained from the PubChem database (ID, 7019) and fully optimized by the MOPAC program (51). The DNA (T7, CG, G10) (52) structure was constructed and optimized by PyMOL, while the crystal structure of double-stranded DNA (dsDNA) (PDB accession no. 4U8A) (35) was downloaded from the PDB database. Then, 9-AA docked with T7, CG, and G10 of DNA and dsDNA (PDB accession no. 4U8A) strands using AutoDock Tools v.1.5.6 (53). Energy minimization was conducted with an Amber14 force field to remove unreasonable atom contacts (54).

**Scanning electron microscopy.** The bacterial strain of *K. pneumoniae* ATCC 700603 was grown in LB medium and incubated at 37°C 180 rpm. The log-phase cultures were washed twice with sterile saline and diluted to a final concentration of a 0.5 McFarland standard in fresh LB medium in the presence of 8× MIC of 9-AA. The bacterial suspension with 1% DMSO treatment was used as a control. After incubation for 1 h at 37°C and 180 rpm, the bacterial suspensions were centrifuged at 40,00 rpm for 8 min, and the bacterial pellets were further washed with PBS and fixed after overnight incubation at 4°C with 2.5% glutaraldehyde. After washing twice with PBS, the samples were continuously treated with a graded ethanol series (30, 50, 70, 80, 90, and 100%). After dehydration, the samples were sputter coated with gold for observation using SEM (Hitachi, Japan).

**Transmission electron microscopy.** Overnight cultures of *K. pneumoniae* ATCC 700603 were diluted 1:100 in fresh LB and grown to the exponential-growth phase. After washing and resuspension in saline solution, the suspension of bacteria was incubated with 8× MIC 9-AA for an hour, and 1% DMSO treatment served as a control. Suspensions of the control and experimental groups were centrifuged (4000 rpm, 8 min), and the supernatants were removed. Specimens were fixed in 0.2 M sodium cacodylate buffer with 2.5% paraformaldehyde, 5% glutaraldehyde, and 0.06% picric acid. Fixed cells were washed with 0.1 M cacodylate buffer and postfixed with 1% osmium tetroxide ($OsO_4$) and 1.5% potassium ferrocyanide [$K_4Fe(CN)_6$] for 60 min. Then, the samples were washed three times with maleate buffer and treated with 1% uranyl acetate in maleate buffer for another 60 min. The bacterial cells were dehydrated through a series of ethyl alcohol (30, 50, 70, 90, and 100% alcohol). Dehydrated specimens were incubated in propylene oxide for 1 h, infiltrated with a 1:1 (vol/vol) solution of propylene oxide and Epon for 1 h, and polymerized at 60°C for 48 h. Ultrathin sections were cut using an ultramicrotome (Reichert Ultracut S). Ultrathin sections (60 nm) were stained with lead citrate. Finally, the cells were observed using TEM (Hitachi, Japan).

**Proton motive force detection by a pH-sensitive probe.** Overnight cultures of *K. pneumoniae* ATCC 700603 were washed and diluted with PBS to an $OD_{630}$ value of 0.1. The suspensions were incubated for 30 min in darkness at 37°C upon the addition of various concentrations of 9-AA and 10 $\mu$M of the pH-sensitive dye BCECF-AM. The fluorescence at excitation and emission wavelengths of 500 and 520 nm, respectively, was determined using a microplate reader (55).

**PMF detection by a membrane potential-sensitive probe.** Exponential-phase *K. pneumoniae* ATCC 700603 was washed three times and adjusted to an $OD_{630}$ of 0.05 in HEPES buffer containing 5 mM glucose and 100 mM KCl (pH 7.2). Then, the bacterial suspension was incubated with 2 $\mu$M $DiSC_3(5)$ in the dark for 30 min. Next, 90 $\mu$L of bacterial suspension and 10 $\mu$L of 9-AA with the indicated concentrations were added to 96-well black-walled plates. Melittin was used as a positive control. Fluorescence (622 nm excitation, 670 nm emission) was detected using a fluorescence microplate reader (32).

**Determination of intracellular ATP levels in bacteria.** An enhanced ATP determination kit (Beyotime, Shanghai, China) was used to detect the intracellular ATP concentration according to the manufacturer's instructions. Briefly, overnight *K. pneumoniae* ATCC 700603 cultures were centrifuged, washed three times, and diluted in PBS to yield an $OD_{630}$ of 0.5. Subsequently, the bacterial suspension was incubated for 1 h at 37°C with the indicated concentration of 9-AA. PBS without DMSO was used as the negative control. After centrifugation at 10,000 rpm and 4°C for 5 min, lysozyme was used to lyse the precipitates. The suspension obtained from the lysis of the pellet was then centrifuged further, and the supernatants were mixed with the working reagents to measure the intracellular ATP level. Following incubation for 5 min at room temperature, the luminescence intensities of mixed samples were assessed by a microplate reader (56).

**Effects of pH variation on the antimicrobial activity of 9-AA.** To study the effect of altering the external pH on 9-AA antibacterial activity, the MICs of 9-AA against *K. pneumoniae* ATCC 700603 at different pH values were tested. NaOH and HCl were utilized to adjust the broth pH value. MIC determination was performed as described above (32).

**Measurement of total reactive oxygen species levels.** ROS levels in *K. pneumoniae* ATCC 700603 treated with 9-AA were determined using 2′,7′-dichlorofluorescein diacetate (DCFH-DA; Beyotime, Shanghai, China) in accordance with the manufacturer's instructions. After incubation with 10 $\mu$M DCFH-DA for 30 min, the bacterial suspension was washed and suspended in PBS. Next, 190 $\mu$L of probe-labeled bacterial cells

was mixed with 10 $\mu$L of different concentrations of 9-AA in a 96-well plate, and polymyxin B (PB) was used as the positive control. The fluorescence intensities were detected at an excitation wavelength of 488 nm and an emission wavelength of 525 nm after another 30 min of incubation at 30℃ (56).

**Surface motility.** Plates containing 0.3% agar and 8 $\mu$g/mL 9-AA were used to determine bacterial motility as previously described with some modifications (57). Briefly, *K. pneumoniae* ATCC 700603 was grown overnight at 200 rpm at 37℃. Bacterial cultures were diluted to an $OD_{630}$ of 0.5, and 2 $\mu$L of the diluted bacterial solution was inoculated on the center of the plate. The migration zone was measured, and images were captured after incubation for 48 h.

**Erythrocyte hemolysis assays.** Human red blood cells (RBCs) were purchased from Hemo Pharmaceutical & Biological Co. (Shanghai, China). Samples were centrifuged at 10,000 rpm for 5 min at 4℃ and resuspended in 1× PBS. Then, the cells were resuspended in PBS mixed with equal volumes of the indicated concentrations of 9-AA and incubated at 37℃ for 1 h, while the 0.5% DMSO- and 0.1% Triton X-100-treated groups were used as negative and positive controls, respectively. Next, the cell supernatants were transferred to a 96-well plate, and the absorbance at 570 nm ($A_{570}$) was detected (58). Meanwhile, RBC pellets were placed on glass slides and observed with CLSM. The hemolysis rate was calculated using the equation:

$$\text{hemolysis } (\%) = \frac{A_{\text{sample}} - A_{0.5\% \text{ DMSO}}}{A_{\text{TritonX}-100} - A_{0.5\% \text{ DMSO}}} \times 100\%$$

**Characterization of 9-AA liposomes.** Liposomes (L) were prepared as follows. Soy phosphatidylcholine (SPC), cholesterol, and polyethylene glycol (PEG) (molar ratio, 80:15:5) were dissolved in chloroform. 9-AA was dissolved in methanol. The above chloroform and methanol were mixed at a ratio of 2:1 (vol/vol) and 9-AA/SPC at a ratio of 3:100 (wt/wt). The organic phase was removed in a rotary evaporator (Yiheng, Shanghai) to obtain a thin lipid film. Then, the thin lipid film was subsequently extruded through 400-, 200-, and 100-nm films to obtain 9-AA(L). 9-AA plus RIF(L), liposomes coloaded with 9-AA and RIF, were obtained as described above and made a ratio of 9-AA/RIF of 1:1 (wt/wt). The final concentrations of 9-AA(L) and 9-AA plus RIF(L) are 1 mg/mL and 0.5 mg/mL plus 0.5 mg/mL, respectively. The zeta potential and particle size of liposomes were analyzed when they were produced. The morphology and fluorescence properties of 9-AA (L) were observed by TEM and CLSM, respectively, as described above. The UV-Vis (UV-visible) absorption and transmission spectra were collected using a PerkinElmer Lambda 1050 spectrometer. Liposome characterization is shown in Fig. S4. The schematic in Fig. S4A indicates that 9-AA encapsulated in liposomes is released more slowly and moderately, hence prolonging its curative effect and lowering its toxicity. 9-AA(L) was spherical in shape and well dispersed, as shown in the TEM images (Fig. S4C). CLSM images showed that liposomeization did not influence its fluorescence characteristics (Fig. S4B). The intensity distribution data and the zeta potentials of 9-AA(L) and 9-AA plus RIF(L) indicated liposomes had a homogenous size distribution and physical stability of colloidal suspension (Fig. S4D to G). The fluorescence properties were unaltered for both 9-AA(L) and 9-AA plus RIF(L), which indicated the structural and fluorescence stability of 9-AA both in the free and liposomal states (Fig. S4H and I).

**Cytotoxicity by Cell Counting Kit-8.** To determine the cytotoxicity of 9-AA and 9-AA(L), the viability of human normal hepatocyte cells (LO2), human hepatocarcinoma cells (HepG2), human kidney proximal tubule epithelial cells (HK2), human renal cell adenocarcinoma (786-O), human microglial clone 3 (HMC3) and human glioblastoma cells (U251) was estimated using CCK-8. U251 and HepG2 cells were cultured in Dulbecco's modified Eagle's medium (DMEM) containing 10% fetal bovine serum (FBS), and LO2 and 786-O cells were cultured in RPMI 1640 medium supplemented with 10% FBS. HK2 and HMC3 cells were cultured in minimum essential medium (MEM) with 10% FBS. One hundred microliters of log-phase cells was seeded into a microplate at a final density of $1 \times 10^4$ per well. Following 24 h of incubation and attachment at 37℃ with 5% $CO_2$, cells were treated with various concentrations of 9-AA and 9-AA(L), and cells treated with 0.1% DMSO were included as a control. Then, CCK-8 reagent was added to each well and incubated for another 4 h. Finally, the absorbance was recorded at 450 nm ($A_{450}$). The $IC_{50}$ was calculated using GraphPad Prism 8.0 software and the cell viability (%) was calculated by the following equation:

$$\text{cell viability}(\%) = \frac{A_{\text{sample}}}{A_{0.1\% \text{ DMSO}}} \times 100\%$$

**In vivo antimicrobial activity.** To explore the bactericidal effect *in vivo*, the wound infection model and skin subcutaneous abscess model were performed in 7-week-old ICR female mice (Hunan SJA Laboratory Animal Co., Ltd., China). For the wound infection model, first, mice were anesthetized with isoflurane by subcutaneous injection, the hair of the dorsal skin was removed, and a wound was cut out using a 6-mm puncher (Φ = 6 mm). The wound site was infected with 50 $\mu$L ($3 \times 10^8$ CFU/mL) *K. pneumoniae* ATCC 700603 suspension. Two hours postinfection, Glaxal Base moisturizing cream containing 0.5% (wt/wt) 9-AA or 0.5% (wt/wt) RIF alone or in combination was applied to the wound. Meanwhile, saline with 1.0% DMSO was applied to the wound as a vehicle control. Each wound was covered with parafilm to prevent ointment leakage and other bacterial infections. Twenty-four hours later, the ointment-containing drugs and the parafilm were replaced. A further 24 h later, the mice were sacrificed, and the wound skin was excised and homogenized with saline. The number of wound bacteria was quantified by dilution plate counting. For the subcutaneous abscess model, mice were anesthetized with isoflurane, and the hair of the dorsal skin was removed. A *K. pneumoniae* ATCC 700603 suspension (50 $\mu$L, $9 \times 10^8$ CFU/mL) was subcutaneously injected into the mouse dorsum. One hour after inoculation, the indicated concentrations of drugs were injected subcutaneously into the infected site. After treatment for 24 h, the mice were euthanized, and the bactericidal effects of the drugs were measured by the standard plate-counting method.

For histopathological analysis, infected skin tissue sections were analyzed via hematoxylin and eosin (H&E) staining. The flowchart of the *in vivo* experiment was plotted using FigDraw.

**In vivo toxicology.** Female ICR mice were randomly divided into 3 groups (5 mice per group). Three groups of mice were administered 1% DMSO, 15 mg/kg 9-AA, and 15 mg/kg 9-AA (L) subcutaneously. Twenty-four hours after subcutaneous drug injection, the mice were euthanized, and whole blood was collected from the orbital vein. Routine blood and organic function biomarker levels were determined. The major organs (heart, liver, spleen, lung, and kidney) were excised and stained with H&E for histopathological analysis.

**Statistical analyses.** All experiments were performed at least in triplicate. The data are expressed as the mean ± standard deviation (SD) and were analyzed using Prism 8.0 (GraphPad Software, San Diego, CA, USA). Statistical significance was determined using unpaired Student's *t* test or one-way ANOVA, and a *P* value of <0.05 was considered statistically significant.

**Ethics approval and consent to participate.** All studies involving animals were conducted in accordance with the Guide for the Care and Use of Laboratory Animals and approved by the Ethics Committee of the Third Xiangya Hospital of Central South University (ID, 2021sydw0245).

## SUPPLEMENTAL MATERIAL

Supplemental material is available online only.

**SUPPLEMENTAL FILE 1**, PDF file, 2.2 MB.
**SUPPLEMENTAL FILE 2**, XLSX file, 0.2 MB.

## ACKNOWLEDGMENTS

This study was supported by the National Natural Science Foundation of China (grant no: 82072350 and 82202591), the Natural Science Foundations of Hunan Province (grant no: 2021JJ40944 and 2022JJ70046), the Key Research and Development Program of Hunan Province of China (grant no: 2022SK2116) and the Project of Scientific Research Plan of Hunan Provincial Health Commission (grant no: B202311000022).

P.S. and Y.W. designed research; Y.L., P.S., Z.L., and S.L. performed the study; P.S., Y.L., and Y.W. analyzed data. All authors wrote and edited the paper.

We declare that we have no competing interests.

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
