## [Reviewer comments · Microbiology Spectrum]

Microbiology Spectrum

Repurposing 9-aminoacridine as an adjuvant enhances the antimicrobial effects of rifampin against multidrug resistant *Klebsiella pneumoniae*

Pengfei She, Yimin Li, Zehao Li, Shasha Liu, Yifan Yang, Linhui Li, Linying Zhou, and Yong Wu

Corresponding Author(s): Yong Wu, The Affiliated Changsha Hospital of Xiangya School of Medicine, Central South University

Review Timeline:

Submission Date:	November 3, 2022
Editorial Decision:	February 17, 2023
Revision Received:	March 8, 2023
Accepted:	March 22, 2023

Editor: Krisztina Papp-Wallace

Reviewer(s): Disclosure of reviewer identity is with reference to reviewer comments included in decision letter(s). The following individuals involved in review of your submission have agreed to reveal their identity: Mikael Young (Reviewer #2)

Transaction Report:

DOI: <https://doi.org/10.1128/spectrum.04474-22>

February 17, 2023

Prof. Yong Wu
The Affiliated Changsha Hospital of Xiangya School of Medicine, Central South University
No.311, Yingpan Road
Changsha
China

Re: Spectrum04474-22 (Repurposing 9-aminoacridine as an adjuvant enhances the antimicrobial effects of rifampin against multidrug resistant *Klebsiella pneumoniae*)

Dear Prof. Yong Wu:

Link Not Available

Sincerely,

Krisztina Papp-Wallace

Journals Department
Reviewer comments:

Reviewer #1 (Comments for the Author):

This manuscript describes characterization of synergistic activity between rifampin and 9-aminoacridine against *Klebsiella pneumoniae*. Synergy was investigated using in vitro assays including the standard checkerboard assay. In vivo experiments demonstrated dose-dependent activity against infection in an animal model.

It is interesting that the 9-AA analogs from figure S7 appear to have greater synergistic activity with rifampin than 9-AA itself. Were any other 9-AA analogs studied? Were any other antimicrobials tested for synergy with 9-AA or analogs to determine if synergy is specific to rifampin?

Line 93: The term "antibacterial growth activity" is unclear. Should this be "inhibition"?

Line 107-108: If MIC was defined based on visible growth what was the OD630 reading used for?

Line 113: I am not aware of a CLSI guideline for checkerboard assays. Please cite.

Line 366-372: Given the potential for DNA interaction of 9-AA, is there concern for more long-term effects (ex. cancer) that cannot be measured in a 24-hour acute toxicity study?

Reviewer #2 (Comments for the Author):

Please proof-read and ensure there are no spelling errors in the paper. While the synergistic activity of 9-AA with RFP is strong, 9-AA displayed only a moderate IC50 with multiple cell lines. It is strongly recommended that the authors look into possibility of a medicinal chemistry development with different analogues of 9-AA to find a version that maintains synergistic activity while being safer for mammalian cells.

Staff Comments:

Preparing Revision Guidelines

Please return the manuscript within 60 days; if you cannot complete the modification within this time period, please contact me. If you do not wish to modify the manuscript and prefer to submit it to another journal, please notify me of your decision immediately so that the manuscript may be formally withdrawn from consideration by Microbiology Spectrum.

Manuscript review

Manuscript # Spectrum04474-22

Submission date : 2022-11-03

Title: Repurposing 9-aminoacridine as an adjuvant enhances the antimicrobial effects of rifampin against multidrug resistant *Klebsiella pneumoniae*

Overall Comments: Pengfei et al have investigated the possibility of repurposing the small molecule 9-aminoacridine (9-AA). They identified 9-AA from a high throughput screening paradigm used to select molecules which display synergistic activity with rifampicin (RFP) against drug resistant *K. pneumoniae*. 9-AA has previous been studied for its potential in cancer and Alzheimer's disease. The manuscript appears to have been well thought out with a solid screening process used to identify the small molecule. In depth studies show the molecule displays a FICI ≤ 0.5 against various strains. Experiments conducted by the authors show that 9-AA exhibits a low probability of resistance development when used together with RFP, it may interfere with the bacteria membrane but CLSM and exogenous DNA studies indicate that DNA maybe its primary target. It displayed a moderate therapeutic window (2 to 4 X MIC) when evaluated with multiple mammalian cell lines. In-vivo analysis showed that the combinational treatment yielded improved activity then either treatment alone. The overall study indicates that 9-AA has antimicrobial activity alone and can be used in combinational therapy against *K. pneumoniae*.

Dear Editor and Reviewers:

Thank you for your letter and for the reviewers' comments concerning our manuscript entitled "Repurposing 9-aminoacridine as an adjuvant enhances the antimicrobial effects of rifampin against multidrug resistant *Klebsiella pneumoniae*." (Spectrum04474-22). Those comments are all valuable and very helpful for revising and improving our paper, as well as the important guiding significance to our researches. We have studied comments carefully and have made correction which we hope meet with approval. Revised portions are marked in highlight in the manuscript. The details of corrections in the text and the related responses are as follows:

Reviewer #1 (Comments for the Author):

This manuscript describes characterization of synergistic activity between rifampin and 9-aminoacridine against *Klebsiella pneumoniae*. Synergy was investigated using in vitro assays including the standard checkerboard assay. In vivo experiments demonstrated dose-dependent activity against infection in an animal model.

1. Comments : It is interesting that the 9-AA analogs from figure S7 appear to have greater synergistic activity with rifampin than 9-AA itself. Were any other 9-AA analogs studied? Were any other antimicrobials tested for synergy with 9-AA or analogs to determine if synergy is specific to rifampin?

Response: We totally understand the reviewer's concern. The time-kill curves of 9-AA analogs were measured and added as a new figure (Figure 7). As shown in figure 7, although the combinations between 9-AA analogs and RFP were efficient in bacterial growth inhibition against *K. pneumoniae* during the 24 h exposure period, no bactericidal activity was observed. However, the 9-AA/RFP combination showed strong bactericidal activity against both of the XDR and PDR strains. In addition, we have supplemented cytotoxicity profile of 9-AA analogs in the Table S3, we found the analogs did not exhibit more safety to the measured human cell lines than 9-AA. Therefore, 9-AA was selected as our main research object.

Meanwhile, the representative checkerboard images of 9-AA and its analogs combined with conventional antibiotics against *K. pneumoniae* were supplemented as Figure S7. However, combining 9-AA and 9-AA analogs with most of the conventional antibiotics had no synergistic effects except for the combination between ethacridine and doxycycline. Our result suggested that the antimicrobial synergistic effects of RFP with 9-AA or 9-AA analogs were antibiotic-specific.

2. Comments : Line 93: The term "antibacterial growth activity" is unclear. Should this be "inhibition"?

Response: We are very sorry for this mistake in our manuscript. We have revised "antibacterial growth activity" to "growth-inhibition activity".

3. Comments : Line 107-108: If MIC was defined based on visible growth what was the OD630 reading used for?

Response: We totally understand the reviewer's concern. The original text of MIC definition in CLSI is as follows: "minimal inhibitory concentration (MIC) – the lowest concentration of an antimicrobial agent that prevents visible growth of a microorganism in an agar or broth dilution susceptibility test"^[1]. Nevertheless, the MIC values can also be established by naked eye analysis and spectrophotometric measurement (OD_{630 nm})^[2,3]. At present, spectrophotometric measurement is more widely used because it is more accurate and easier to standardize. To avoid confusion, we have revised "visible growth" to "measurable bacterial growth" in the new manuscript.

[1] CLSI. 2022. M23. Methods for Dilution Antimicrobial Susceptibility Tests for Bacteria That Grow Aerobically, 11th ed Clinical and Laboratory Standards Institute, USA. <https://www.clsi.org/standards/products/microbiology/documents/m23/>.

[2] Aldape MJ, Bayer CR, Rice SN, Bryant AE, Stevens DL. Comparative efficacy of antibiotics in treating experimental Clostridium septicum infection. *Int J Antimicrob Agents*. 2018;52(4):469-473. doi:10.1016/j.ijantimicag.2018.07.009

[3] Song M, Liu Y, Huang X, et al. A broad-spectrum antibiotic adjuvant reverses multidrug-resistant Gram-negative pathogens. *Nat Microbiol*. 2020;5(8):1040-1050. doi:10.1038/s41564-020-0723-z

Spohn R, Daruka L, Lázár V, et al. Integrated evolutionary analysis reveals antimicrobial peptides with limited resistance. *Nat Commun*. 2019;10(1):4538. Published 2019 Oct 4. doi:10.1038/s41467-019-12364-6

4.Comments : Line 113: I am not aware of a CLSI guideline for checkerboard assays. Please cite.

Response: We are sorry for our careless, we have cited it in the new manuscript.

5.Comments : Line 366-372: Given the potential for DNA interaction of 9-AA, is there concern for more long-term effects (ex. cancer) that cannot be measured in a 24-hour acute toxicity study?

Response:

We totally understand reviewer's worry about long-term toxicity. Firstly, in the DNA-9AA interaction related mechanistic experiments, no obvious infiltration was observed in mammalian cells by 9-AA, while the bacterial cell membrane and DNA were targeted by 9-AA (Figure 3A), which indicates 9-AA probably selectively targets bacteria but not mammalian cells in a short time treatment. Secondly, this study focus on localized infection model and topical medication, it is not easy to produce systemic and long-term toxicity. Thirdly, drug combination can lower the drug concentrations and thus reduce potential toxicity including cytotoxicity and genotoxicity. Finally, the clinical course of an antibiotic usage is often shorter than two weeks, otherwise it will cause flora imbalance and drug resistance^[1]. Besides, a previous study has shown low genotoxicity of 9-AA^[2]. In addition, we will continue conduct further structural modifications of 9-AA to improve its efficacy and safety. We have added relevant discussion in the revised manuscript. Please see line 570-581 in marked up manuscript.

[1] Kalil, Andre C et al. "Management of Adults With Hospital-acquired and Ventilator-associated Pneumonia: 2016 Clinical Practice Guidelines by the Infectious Diseases Society of America and the American Thoracic Society." *Clinical infectious diseases: an official publication of the*

Infectious Diseases Society of America vol. 63,5 (2016): e61-e111. doi:10.1093/cid/ciw353

[2] Wilson WR, Harris NM, Ferguson LR. Comparison of the mutagenic and clastogenic activity of amsacrine and other DNA-intercalating drugs in cultured V79 Chinese hamster cells. *Cancer Res.* 1984 Oct;44(10):4420-31. PMID: 6547875.

Reviewer #2 (Comments for the Author):

Please proof-read and ensure there are no spelling errors in the paper. While the synergistic activity of 9-AA with RFP is strong, 9-AA displayed only a moderate IC50 with multiple cell lines. It is strongly recommended that the authors look into possibility of a medicinal chemistry development with different analogues of 9-AA to find a version that maintains synergistic activity while being safer for mammalian cells.

Response: Among the analogs of 9-AA, euflavine and ethacridine had synergistic antibacterial effect with RFP, however these synergisms are weaker than 9-AA (Figure 1 and Figure 7). Meanwhile, except for one cell, the analogs were more toxic to most of the measured human cell lines than 9-AA (Table S3). Therefore, 9-AA was selected as our main research object. As for reviewer's concern about cytotoxicity of 9-AA, we have several explanations. Firstly, drug combination can lower the drug concentrations and thus reduce potential toxicity including cytotoxicity and genotoxicity. Secondly, 9-AA and 9-AA (L) at effective concentration did not cause skin redness, ulceration, bleeding or histological alterations in wound infection model and skin subcutaneous abscess model (Figure 6B, D, F). Similarly, no statistical significance was detected for hematological liver functional, renal functional, or myocardial functional biomarkers between the drug-treated and vehicle groups (Figure S6A, B). The H&E staining of these organs with 9-AA and 9-AA (L) treatment also showed no pathological changes (Figure S6C). In summary, though 9-AA displayed only a moderate IC50 with multiple cell lines, 9-AA is still a promising antimicrobial agent with low toxic effects in localized infection model as an adjuvant. 9-AA (L) further improved the safety of 9-AA and retained antibacterial activity. To further improve antibacterial activity and reduce toxicity of 9-AA, we will conduct further structural modifications of 9-AA in the future.

[1] Wilson WR, Harris NM, Ferguson LR. Comparison of the mutagenic and clastogenic activity of amsacrine and other DNA-intercalating drugs in cultured V79 Chinese hamster cells. *Cancer Res.* 1984 Oct;44(10):4420-31. PMID: 6547875.

March 22, 2023

Prof. Yong Wu
The Affiliated Changsha Hospital of Xiangya School of Medicine, Central South University
No.311, Yingpan Road
Changsha
China

Re: Spectrum04474-22R1 (Repurposing 9-aminoacridine as an adjuvant enhances the antimicrobial effects of rifampin against multidrug resistant *Klebsiella pneumoniae*)

Dear Prof. Yong Wu:

Your manuscript has been accepted, and I am forwarding it to the ASM Journals Department for publication. You will be notified when your proofs are ready to be viewed.

Sincerely,

Krisztina Papp-Wallace
Editor, Microbiology Spectrum
